# Beneficial Effect of the New *Leptodophora* sp. Strain on Development of Blueberry Microclones in the Process of Their Adaptation

**DOI:** 10.3390/microorganisms11061406

**Published:** 2023-05-26

**Authors:** Yulia S. Topilina, Evgeniya A. Luk‘yanova, Lubov B. Glukhova, Margarita N. Shurupova, Anna L. Gerasimchuk, Yulia A. Frank, Dmitry V. Antsiferov

**Affiliations:** 1Biological Institute, National Research Tomsk State University, Lenina Ave., 36, 634050 Tomsk, Russia; madam.topilina@mail.ru (Y.S.T.); lubov.b.gluhova@gmail.com (L.B.G.); rita.shurupova@inbox.ru (M.N.S.); gerasimchuk_ann@mail.ru (A.L.G.); yulia.a.frank@gmail.com (Y.A.F.); 2LLC Darwin, Str. Vysockogo Vladimira, 28, 634040 Tomsk, Russia; evgenialukjanova@gmail.com

**Keywords:** ericales, ericoid mycorrhiza, blueberry, microclones, phylogeny

## Abstract

The paper searches for new solutions for the development of highbush blueberry orchards (*Vaccinium corymbosum* L. (1753)) in Western Siberia. All species of the genus *Vaccinium* display special symbiotic mycorrhizal associations with root systems—ericoid mycorrhiza, which essentially enhances the formation of adventitious and lateral roots. For the first time, we obtained pure cultures of micromycetes associated with the roots of wild species of the family *Ericaceae* in the Tomsk region, Russia. With regard to the data of molecular genetic analysis of the ITS region sequence, we selected the BR2-1 isolate based on its morphophysiological traits, which was assigned to the genus *Leptodophora*. Representatives of this genus typically enter into symbiotic relationships with heathers to form ericoid mycorrhizae. We studied the effect of strain BR2-1 on the development of microclones of the highbush blueberry var. Nord blue during their in vitro adaptation and showed its beneficial effect on growth and shoot formation in young plants. Experiments performed using submerged and solid-state methods showed that the most optimal method for commercial production of BR2-1 is cultivation on grain sterilized by boiling, followed by spore washing.

## 1. Introduction

Plants of the genus *Vaccinium* (family *Ericaceae*) cover extensive Holarctic ranges with significant disjunctions in the majority of the northern hemisphere. The species grow in North America (Western Canada, Northwest and Southwest USA—Rocky Mountains), Greenland, almost all of Europe (except Greece), the European part of Russia, the Urals, the Caucasus (Georgia and Armenia), Turkey, Western and Eastern Siberia, Mongolia, and in Japan [1]. In nature, the species grow in coniferous and mixed moist or swamp forests and in the tundra; in the highlands, they extend to subalpine meadows. The soil in their habitats is rich in organic matter, but its acidity sufficiently decreases its nutrient availability for plant growth [2]. The prevalence of heathers in these conditions is due to their special symbiotic relationships with some soil micromycetes, referred to as ericoid mycorrhizae. When a hypha comes into contact with the plant roots, it spreads through the intercellular space of the root hair and penetrates into the epidermal cell, forming an arbuscula (dichotomously branched hyphae of complex shapes that penetrate into parenchymal cells of the root). In arbuscules, the exchange of metabolites between mycorrhiza components is the most intensive. The ericoid mycorrhiza permits the fungus to obtain polysaccharides from the plant, while the fungus helps the plant to receive water, organic and mineral forms of nitrogen and phosphorus [2]. In addition, fungi can bind and remove zinc, copper and other heavy metals, which are toxic to the plant in large quantities [2]. Increased drought tolerance was shown for *Vaccinium myrtilloides* blueberry plants inoculated with four ericoid mycorrhizal (ErM) fungi—*Pezicula ericae*, *Pezoloma ericae*, *Meliniomyces variabilis*, and *Oidiodendron maius* isolated from the roots of plants of this genus [3]. Mycorrhizal fungi, including ericoid fungi, promote adventitious and lateral root formation. The benefit from plant–fungus symbiosis is maximal if the inoculation occurs during adventitious or lateral root formation, that is, at the earliest stage of plant development. Mycorrhizae significantly increase the root and shoot biomass of blueberries and lingonberries, and hence, insure a high yield of these plants [4]. However, the ericoid mycorrhiza is the least studied among all the known forms of mycorrhizae [5]. This may be related to the paraphyletic origin of ErM species. Early attempts to identify fungi of this physiological group were not successful because they were based on the study of their morphophysiological traits. Nevertheless, these methods enabled the classification of fungi from other groups of a monophyletic origin, for example, arbuscular mycorrhiza-forming fungi [6]. The diversity of the fungal community associated with the roots of the studied plants, which can include different pathogens and symbionts, hampers investigation. Therefore, confirmation of the ability to form ErM mycorrhiza is required for each isolate obtained. In recent years, strain identification and classification have been greatly simplified due to advances in molecular genetic methods and the creation of databases. However, the ability of new strains to form ErM mycorrhiza must be confirmed experimentally. For example, among numerous species phylogenetically close to *Rhizoscyphus ericae*, only two are found on the roots of heather plants, while others are associated with the roots of plants from other families [7]. In addition, the biogeography of ErM fungi has not been studied at all. There are comprehensive data on the distribution of host plants, yet it is not possible to associate their habitats with symbionts, because species-specificity is not reported for ErM fungi. Most of the studies on ErM fungi were conducted in the northern hemisphere, mainly in North America and northwestern Europe, and in Japan. We did not find any data reported for ErM species from the territory of the Russian Federation and neighboring countries. An increasing number of molecular studies enables us to obtain indirect data on the distribution of ErM fungi based on the results of soil microbiota diversity, because they are often associated with the roots of plants from other families [8].

Most heather endomycorrhiza isolates belong to the division *Ascomycota* (up to 84 percent of all species associated with different parts of the plant [9]), and they are mainly represented by two orders: *Helotiales* and *Onygenales*. Recent studies have also shown the involvement of basidiomycetes in symbiotic relationships with plants of the family *Ericaceae* [10,11,12]; the proportion of its representatives in the community varied from 7 percent in flowers to 27 percent in stems [9]. Among ericoid mycorrhizae, *Rhizoscyphus ericae* species are the most common [7,13,14]. This group includes both ectomycorrhizal and endomycorrhizal species with the identity of the ITS region not less than 84 percent [6], including species of the genus *Cadophora* [15]. *Cadophora* species are the most common among the species of the order *Helotiales* [16,17]; they exhibit pigmented hyphae and slow mycelium growth [18]. A number of *Cadophora* species are involved in ericoid associations with plants of the genus *Vaccinium* [6] and form ectomycorrhizae on the roots of *Pinus strobus* L. [19]. However, many species are referred to the genus *Cadophora* based only on their morphophysiological traits. For example, until recently, *C. orchidicola* sp. had no particular position within the paraphyletic genus *Cadophora*, due to the absence of its molecular markers. At present, the phylogenetic analysis of the four marker sequences ITS, 28S rDNA, RPB2, and EF1-alpha obtained from the ex-type strain of *C. orchidicola* has allowed the isolation of a new genus *Leptodophora*, with the type strain *L. orchidicola* UAMH5422, and inclusion in the genus of three new, closely related type species *L. gamsii* CBS146379, *L. echinata* CBS146383 and *L. variabilis* CBS146360. In addition, to stabilize the *Cadophora* taxonomy, a new genus *Collembolispora* was proposed, with the type strain *C. dissimilis* CBS146372, which produces phialide conidiogenic cells, and which was previously regarded as *Cadophora* sp. [20]. The position of the newly proposed genera *Leptodophora* and *Collembolispora* in the order *Helotiales* has not yet been determined, and the phylogeny of this order is to be reconsidered in the future.

A recent study of *Vaccinium myrtillus* plants has shown that ErM fungi can colonize not only the roots, but also stems and leaves [9]. This feature was revealed in fungi of the genus *Hyaloscypha*. Colonization of the above-ground organs can be attributed to the evolutionary similarity between the representatives of this genus and non-mycorrhizal fungal endophytes. However, the role of this interaction in the above-ground organs of heather plants is still unknown.

One of the most widespread commercial crops among species of the genus *Vaccinium* is highbush blueberry (*Vaccinium corymbosum* L. (1753)), which is used as a fruit and ornamental plant. For the mass production of fruit and berry seedlings, including highbush blueberry, the micropropagation method is used [21]. Microclones are obtained in sterile conditions; therefore they lack symbionts, and many microplants die during their adaptation to ex vitro conditions. Preliminary microclone root inoculation with spores of symbiotic micromycete prior to adaptation can increase the plants’ stress resistance and growth rate [22].

The aim of this study was to identify and isolate endomycorrhizal micromycetes from the roots of plants of the genus *Vaccinium*, and to investigate their effect on the adaptation, growth and development of microclones of garden blueberry var. Nord blue.

## 2. Materials and Methods

### 2.1. Collection of Primary Hosts and Identification of Mycorrhiza

For the identification of mycorrhiza-forming fungi and isolation of the pure culture, we used the roots of lingonberry (*Vaccinium vitis-idaea*) and blueberry (*Vaccinium myrtillus*) plants collected in the forest near the village of Takhtamyshevo (56.383895 N 84.859805 E), Tomsk region (Western Siberia, RF). The plant roots were dug out together with a clod of earth, which was then placed in an individual clean plastic bag for transportation.

To identify mycorrhizae and determine the extent of colonization, the roots of the collected plants were washed in running water; then, the roots were discolored in a water bath in hydrogen peroxide solution and stained with lactophenol-trypan blue solution [23,24,25]. Slides of the stained roots were studied using a Biomed-6 microscope equipped with a KF-4 phase-contrast device (Russia, Moscow) and a ToupCam UCMOS05100KPA camera (Hangzhou, China). The extent of root colonization was determined from fifty visual fields, as described by Smith and Dickson [24].

A similar procedure was used to identify mycorrhizae on microclonal blueberry plants inoculated with spore suspension during adaptation.

### 2.2. Obtaining Pure Cultures of Micromycetes

We rinsed 2–3 cm long root fragments with 10 percent Domestos solution for 1 min and then washed them in running water for 30 min. After that, the roots were stored in 70 percent ethanol solution in a laminar box for 1 min, and then they were immersed in a SYNERGETIC non-chlorine bleach solution (1:1) for 25 min [26]. After sterilization, the roots were washed five times in sterile distilled water and cut into 5 mm fragments. Then, the roots were placed in Petri dishes containing Czapek-Dox agar medium modified with 1 g/L yeast extract and with added antibiotics: 300 mg/L cefotaxime and 100 mg/L chloramphenicol [27]. The plates were incubated in the dark at 24 °C for 7 d. The grown mycelium was separated with a needle, and then transferred to Petri dishes with similar medium without antibiotics. For further study, we employed pure culture, which corresponded to the morphology of mycorrhizal symbiont fungi of the family *Ericaceae* described in the literature.

### 2.3. Molecular Identification

Mycelium DNA was extracted using the phenol-chloroform method described by Joseph Sambrook and David W. Russell [28], with liquid nitrogen pretreatment. The extracted DNA was stored at −20 °C until further use. The ITS region was used as a molecular marker [29], which was amplified using a pair of primers ITS1 and ITS4 [30]. Amplification was performed using a thermal cycler (Mastercycler nexus X2, Eppendorf, Hamburg, Germany) with 40 µL of the PCR mixture containing 4 µL of 10× buffer (Biolabmix); 2.5 mM MgCl_2_ (Biolabmix); 0.2 mM each of dNTPs (Biolabmix); 25 pmol mL${}^{-1}$ of each primer (Synthol); 1 unit of HS-Taq polymerase (Biolabmix); and 60–100 ng of genomic DNA. The amplification program was as follows: initial denaturation at 95 °C for 2 min; then 35 cycles, including denaturation at 95 °C for 30 s, primer annealing at 55 °C for 30 s, and elongation at 72 °C for 1 min; final elongation at 72 °C for 10 min [31]. The PCR results and the length of the obtained fragments (600 bp) were evaluated using electrophoresis in 1 percent agarose gel using SYBR Green I [32]. Commercial nucleotide sequencing with enzymatic purification was performed at CJSC Syntol (Moscow, Russia).

The processed and assembled sequences were compared with the ITS regions available in the GenBank databases using the BLASTn on the NCBI website www.ncbi.nlm.nih.gov [33] (accessed on 21 March 2023). The closest matches (more than 95 percent homology) obtained from the GenBank database were used for cluster alignment and phylogenetic analysis. All sequences were aligned using the Clustal X algorithm with default settings [34] and the U-GENE software v33.0 [35]. Phylogenetic analysis was performed following the neighbor-joining method using the MEGA 11 software [36]. The reliability of the phylogeny was tested via bootstrap (1000 iterations).

### 2.4. Cultivation on Different Substrates

Different methods used in the commercial production of fungal biological preparations, namely, submerged and solid-state methods, were employed to cultivate a strain for assessing its potential for commercial production.

Submerged cultivation to increase biomass was performed in 100 mL of Czapek-Dox liquid medium on a shaker (100 rpm) in 250 mL Erlenmeyer flasks. Agar blocks 5 × 5 mm in size, 3 fragments per flask, were used as an inoculum. Cultivation continued until the mycelium attained 2/3 of the volume of the culture liquid.

Solid-state cultivation was performed on a grain substrate in 500 mL Erlenmeyer flasks. The substrate was prepared as follows: wheat grain was boiled in distilled water for 30 min, then the water was drained, and an even layer of grain was laid out on a cloth and dried in air until completely cooled. The cooled grain was mixed with chalk and gypsum added in a weight ratio of 3 percent and 1 percent, respectively. Flasks, filled with the resulting substrate to 2/3 of the volume and closed with cotton plugs, were autoclaved at 121 °C for 25 min [27]. A 14-day-old mycelium grown in agar medium was used as an inoculum. The inoculum was added in the amount of 1/2 Petri dish per flask. The flasks were incubated at 24 °C until complete fouling of the substrate and active sporulation. Then, the spores and mycelium were washed off the grain with water mixed with 0.1 percent TWEEN-80.

### 2.5. Adaptation of Microclones of Sterile Host Plants

Experimental plants were obtained through micropropagation at Future Flora Lab, Moscow, RF. A total of 60 developed highbush blueberry microclones (*Vaccinium corymbosum* L.) Nord blue var. were washed off the agar and planted in the moistened substrate in the prepared greenhouses. The substrate was prepared from Agrobalt-V peat with a pH of 3.0–4.5 (St. Petersburg, RF), perlite, and distilled water in a ratio of 10:10:2 by volume. The prepared substrate was autoclaved at 121 °C for 60 min. The greenhouses were pre-soaked in a solution of hydrogen peroxide (6 percent) for 12 h. The microclones were divided into control and experimental groups of 30 microclones each. Before planting, the roots of the experimental plants were immersed for 10 s in spore wash-off from Petri dishes with BR2-1 colonies; the roots of control plants were left untreated. The greenhouses were hermetically sealed with an adhesive tape and placed on the shelves exposed to lighting of 3500 Lx. The greenhouses were closed for 2 weeks; then, over one week, the humidity was gradually reduced, and finally the greenhouse covers were opened.

### 2.6. Collection and Analysis of Data on Vaccinium corymbosum L. Microclones

After the 3-week adaptation period, three parameters were measured: average shoot length, average number of shoots, and average leaf size. Observations were carried out for 11 weeks after adaptation completion. An increase in the shoot length was estimated by calculating the difference between the shoot length after opening of the greenhouse cover and its length after 11 weeks. The average leaf size was determined using the weight method [37]. The volume of the root system was assessed visually during plant transplantation. The reliability of the obtained results was assessed using the Mann–Whitney test (U-test) [38].

## 3. Results and Discussion

### 3.1. Isolation and Identification of Pure Culture

Arbuscules and mycelium strands in the intercellular space were found in the epidermal layer of the stained root preparations of wild lingonberries and bilberries (Figure 1). Colonization of their epidermal layer attained 41 percent, which is in good agreement with data from the literature. In natural conditions, the root colonization of heather plants typically varies between 0.5 and 44 percent [39], but the colonization of marsh communities can exceed 90 percent [40]. Numerous scientists report that the degree of ErM root colonization by fungi depends on the season, a finding which is associated with active root growth during the vegetation period [41]. Because the studied samples were collected during the dormant period (in autumn, after fruiting), it can be assumed that the degree of ErM root colonization by fungi could be higher in the earlier period.

The above method was employed to isolate more than 20 pure cultures of fungi from the roots of the studied plants. We studied their morphological and physiological traits, which allowed us to initially remove the samples that did not correspond to ErM fungi. One of the isolates was identified as a widespread phytopathogen, *Penicillium thomii*. Based on available data from the literature on macro- and micromorphology with regard to the nature of growth in dense media, we selected only 10 isolates from the total number of obtained pure cultures. All of them showed slow growth, the colonies were of dark color, and the mycelium was pigmented. BR2-1 isolated from lingonberry roots was chosen as the main study object. This strain formed colonies with a smooth white edge and a brown colored middle; the colony reverse and the exudate were black (Figure 2b) [42], and the average daily growth rate in Czapek-Dox medium was 0.4 mm/d. The BR2-1 colonies grown on potato dextrose agar had a wider white margin (Figure 2c), while in maltose–dextrose medium, they formed aerial mycelium (Figure 2a) stained olive brown. The strain hyphae were stained dark gray, were segmented and contained numerous vacuoles (Figure 2d–f). These traits are typical of many nonpathogenic endophytic fungi referred to as ‘dark septate endophytes’ (DSE), because they exhibit melanized mycelium and are mainly associated with plant roots [43]. Phylogenetic analysis of the ITS region sequence of the isolate assigned BR2-1 to the fungal genus Leptodophora. Sequence similarity was identified via two representatives of the genus, *L. echinata* Koukol et Macia-Vicente (98.62 percent homology) and *L. gamsii* Koukol et Macia-Vicente (98.28 percent homology), which were isolated from the roots of Central European plants [44]. These species were previously assigned to the genus *Cadophora*. The genus *Cadophora* is paraphyletic and most of its representatives form endomycorrhizae with various plant species, including heather plants [43]. Based on the analysis of new molecular data, *L. echinata* and *L. gamsii* were segregated into a separate genus [20]. Figure 3 presents a phylogenetic tree indicating the position of *Leptodophora* sp. BR2-1 relative to other fungi that form mycorrhizae with plants of the genus *Ericaceae*. The sequence was deposited in the GenBank database under accession number OQ257002.

### 3.2. Commercial Cultivation

Submerged cultivation and solid-state cultivation were both successful for BR2-1 (Figure 4). However, the substrates were completely fouled within 21 days because the culture growth was slow. According to the experimental results, cultivation on a sterile grain substrate was the most optimal method of commercial production, because submerged cultivation is more economically costly. The final form of the biological preparation can be of three types: fresh fouled grain substrate; spore wash-off from fouled grain substrate supplemented with surfactants and preservatives; or fouled grain substrate ground and dried at low temperature. The first two types require stricter storage and transportation conditions and have a shorter shelf life compared to the third type. The grain substrate can be packaged in glass containers or polypropylene bags supplied with a filter. Both a suspension obtained by washing off a Petri dish and that obtained by small-scale submerged cultivation can be used as a primary inoculum.

### 3.3. Development of Microclones during Adaptation

In the experiment, the average shoot growth in both groups was virtually similar. The average increase in the control and experimental groups attained 2.6 cm and 2.56 cm, respectively, whereas the other two indicators differed. In the control group, the average number of shoots per individual was 2.5 ± 0.2 pcs, and in the experimental group, it was 3.3 ± 0.3 pcs. In the control group, the average leaf plate area was 56 ± 2 mm^2^, and in the experimental group, it was 62 ± 2 mm^2^. Bizabani and Dames [45] showed a positive effect of various ErM fungi on the development of blueberry plants. The authors reported an increased shoot length and total dry weight in an experimental group of sterile blueberry plants inoculated with strains from the genera *Lachnum* and *Cadophora*. In addition, the authors revealed the variety dependence; for example, blueberry plants of the Chandler, Bluecrop and Spartan varieties did not show any changes in the shoot length when inoculated with these strains, and the colonization of the roots of these plants was minimal. Therefore, for commercial use, the variety specificity of each proposed strain must be studied thoroughly.

In our study, we employed the nonparametric Mann–Whitney test to compare independent experimental samples. The analysis showed that the measured parameters such as the increased shoot length and leaf blade area did not show significant differences between the experimental and control samples (*p* = 0.8 and *p* = 0.14, respectively). A significant difference was found only in the number of shoots (*p* = 0.02) (Figure 5). The state of the root system was assessed visually, because the studied plants were of commercial interest, and commonly used methods require their complete drying. In contrast to the control group, plants from the experimental group had a more branched root system with a large number of lateral roots (Figure 6). Root fragments of the adapted plants were examined for the presence of mycorrhizae. Microscopic analysis of stained preparations revealed successful root colonization by *Leptodophora* sp. BR2-1 in microclones from the experimental group (Figure 7). The figure shows mycelium strands in the intercellular space and arbuscules. The degree of colonization of the epidermal layer of roots was 22 percent, which corresponds to the data obtained in a study of the root system of blueberry plants in gardens [39,46]. The authors assume that a low intensity of mycorrhiza colonization may be associated with fertigation of the studied fields.

### 3.4. Effect of BR2-1 Isolate on Agricultural Crops

An aqueous suspension of spores and mycelium fragments of *Leptodophora* sp. BR2-1 used to inoculate highbush blueberry microclones from the experimental group was employed to cultivate the soil and treat the adapted microclones of garden gooseberry (*Ribes uva-crispa* L.) var. Grushenka, garden strawberry (*Fragaria ananassa Duchesne ex Weston*) var. Zenga-Zengana, and sweetberry honeysuckle (*Lonicera caerulea* L.) var. Aurora. After 3-week observations, the treated plants showed no visible differences in their condition (growth inhibition, pigmentation, wilting, yellowing) compared to the controls. Therefore, it can be concluded that this isolate is not a pathogen for the studied agricultural plants.

## 4. Conclusions

Plants of the family Ericaceae grow in acidic soils with a poor mineral composition; therefore, they require specific symbiotic relationships with micromycetes for normal development. We have shown that the degree of symbiont colonization attained 41 percent on the roots of wild lingonberry and blueberry plants, which were used as a source for pure culture isolation. According to data in the literature, the morphophysiological characteristics of the selected isolates corresponded to the mycrysis-forming symbiont fungi of the family Ericaceae. Based on the phylogenetic analysis of one of the molecular markers, BR2-1 was assigned to the genus Leptodophora, and the representatives of this genus were reported to form the ericoid type of mycorrhiza. Laboratory experiments showed that the inoculation of highbush blueberry var. Nord blue with *Leptodophora* sp. BR2-1 stimulated shoot and root formation, the fungus successfully colonized the roots of the experimental plants, and after 14 weeks of the experiment, the degree of colonization attained 22 percent. In this study, pure cultures of endomycorrhizal fungi were obtained for the first time from plants of the family *Ericaceae* growing in the Tomsk region, a method for producing BR2-1 was developed, and a growth-stimulating effect of this strain on shoot formation in blueberry plants was shown. Thus, *Leptodophora* sp. BR2-1 can be used to develop a biopreparation to increase the crop yield of berries from the genus *Vaccinium*.

## Figures and Tables

**Figure 1 microorganisms-11-01406-f001:**
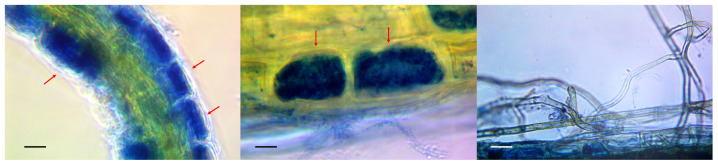
Stained micropreparations indicating endomycorrhizae in the roots of lingonberry plants. Phase contrast microscopy. Arrows show the presence of mycorrhiza formation in the epidermis cells. Scale is 10 µm.

**Figure 2 microorganisms-11-01406-f002:**
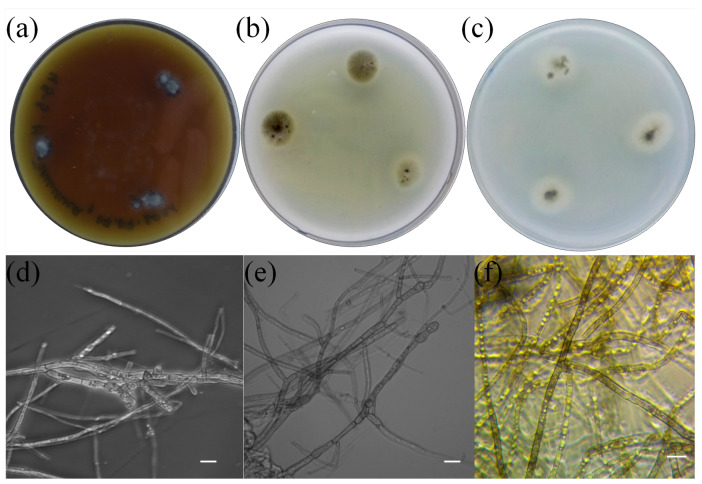
BR2-1 colonies in dense media (Revers): (**a**) maltose–dextrose agar, (**b**) Czapeka-Doxa, (**c**) potato agar and (**d**–**f**) mycelium micromorphology. Phase contrast microscopy. Scale is 10 µm.

**Figure 3 microorganisms-11-01406-f003:**
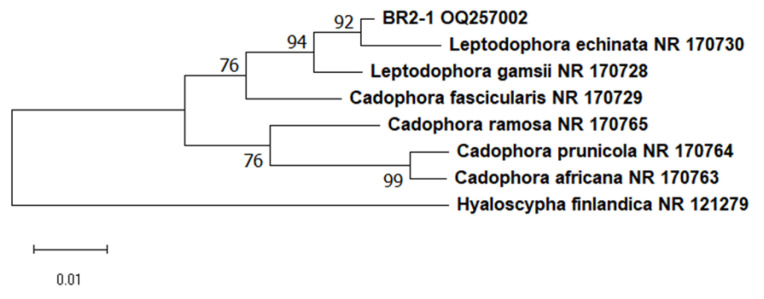
Phylogenetic tree indicating the position of BR2-1 based on the ITS sequence. The tree was built using the neighbor-joining method.

**Figure 4 microorganisms-11-01406-f004:**
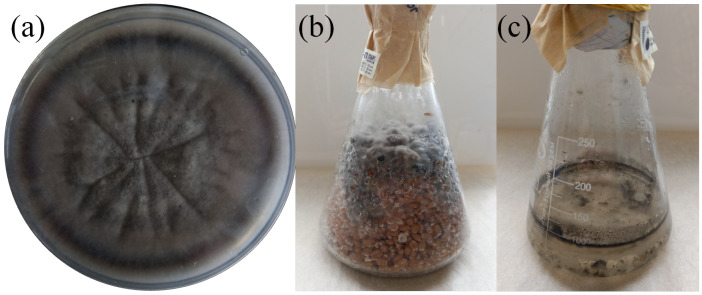
Cultivation of BR2-1 in different media. (**a**) Czapek-Dox agar medium, (**b**) boiled grain (7 d), (**c**) Czapek-Dox liquid medium (7 d).

**Figure 5 microorganisms-11-01406-f005:**
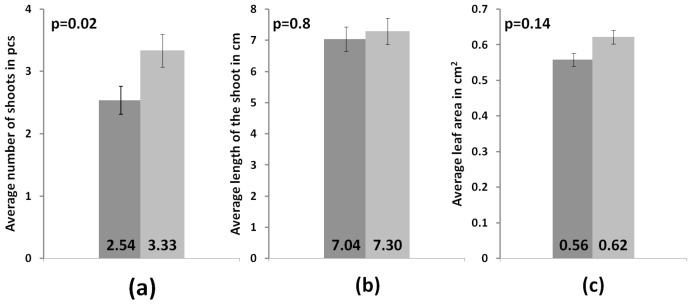
Differences between the control group of microclones (dark gray) and the experimental group (light gray) (**a**) in the average number of shoots, (**b**) in the average length of shoots and (**c**) in the average leaf area.

**Figure 6 microorganisms-11-01406-f006:**
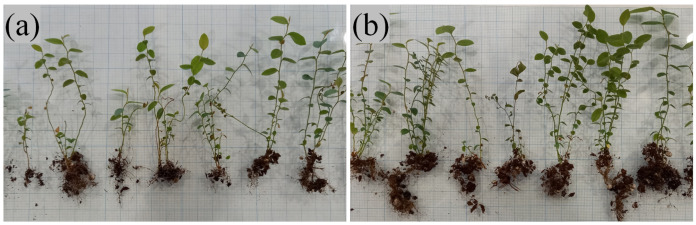
Highbush blueberry microplants var. Nord Country after 3 weeks of adaptation: (**a**) control plants, (**b**) plants inoculated with mycelium and spores of *Leptodophora* sp. BR2-1.

**Figure 7 microorganisms-11-01406-f007:**
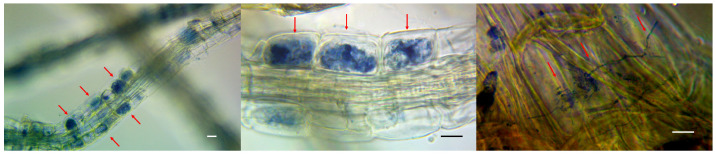
Stained slides showing endomycorrhizae in the roots of blueberry plants from the experimental group inoculated with *Leptodophora* sp. BR2-1. Phase contrast microscopy. Arrows show the presence of mycorrhiza formation in the epidermis cells. Scale is 10 µm.

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
