# Peer review of "Beneficial Effect of the New Leptodophora sp. Strain on Development of Blueberry Microclones in the Process of Their Adaptation"

_microorganisms, 2023, doi:10.3390/microorganisms11061406_

Round 1
Reviewer 1 Report
This manuscript introduces a Leptospira sp. strain isolated from the roots of the genus Vaccinium, which can achieve an ideal commercial production through solid-state methods. Experiments have shown that this strain can colonize the roots of highbush blueburry var. Nord Blue and stimulated the formation of blueberry shoot and root, which has a beneficial impact on the growth and development of blueberry microclones. Plant endophytes have been a research hotspot in recent years and a topic of great interest to the researchers in related fields. This article provides a reference basis for the development of highbush blueberry orchards in Western Siberia.
The article has clear purpose and logic, reasonable complete experimental methods and refined language. The experimental results match the research objectives and the references conform to the subject requirements. However, there still has some small vulnerabilities, and several suggestions are made here:
1. Whether disinfection of sterilized plant roots was verified at the stage of isolation of endophytic fungi to ensure that all resulting strains for subsequent experiments were endophytic fungi. If there has this experiment, please specify.
2. It is suggested to add data charts of blueberry average shoot length, average number of shoots and average leaf size in the results section, in order to more intuitively understand the differences between the experimental group and the control group and increase the article credibility.
The manuscript as a whole is innovative, well-defined, and of high practical value, acceptable after revision.
Author Response
Dear Sir/Madam,
Thank you so much for the appreciation of our work and for the valuable comments.
Comment 1. The routinely used method of surface sterilization of the material was used which are widely applied for in vitro plant cultivation including industrial scales. According to the results of our preliminary research (unpublished), it allows 95% efficiency with no injuring plant tissues compared to using chlorine-containing substances.
All the fungal strains selected and described in the manuscript showed a close homology of the ITS sequence with endophytes (Leptodophora, Umbelopsis, Lachnum, Phialocephala, and Oidiodendron) for which the ability to ERM is documented. This excludes the accidental entry of fungi into the culture and confirms their participation in endosymbiosis.
Someone has an opinion that all isolated endophytes require confirmation of ERM capability by re-infection of sterile plants. We managed to conduct a similar experiment with one of the isolates described in the manuscript.
We thank you for this comment and will take this shortcoming into account in the future work by testing the disinfection effectiveness by imprinting on a rich nutrient medium.
Comment 2. The diagrams were prepared and included in the revised version of the manuscript.

Reviewer 2 Report
This manuscript provides new and exciting data. For the first time, the authors obtained pure cultures of 4 micromycetes associated with the roots of wild species of the family Ericaceae in Tomsk region (Russia).
The paper is primarily well-structured and written. It is easy to follow. The work appears to be extensive and the data sound. The results are revealing and potentially useful. So I support further publication.
Author Response
Dear Sir/Madam,
Thank you for the appreciation of our work. Receiving feedback from a highly qualified specialist is very important for us.